# Comparison of respiratory symptoms and pulmonary functions of adult male cigarette smokers and non-smokers in Sri Lanka; A comparative analytical study

**Prasanna Herath**[1]*, **Savithri Wasundara Wimalasekera**[2], **Thamara Dilhani Amarasekara**[3], **Manoj Sanjeewa Fernando**[4], **Sue Turale**[5]

1 Department of Nursing and Midwifery, Faculty of Allied Health Sciences, General Sir John Kotelawala Defence University, Dehiwala-Mount Lavinia, Sri Lanka, 2 Faculty of Medical Sciences University of Sri Jayewardenepura, Nugegoda, Sri Lanka, 3 Department of Nursing and Midwifery, Faculty of Allied Health Sciences, University of Sri Jayewardenepura, Nugegoda, Sri Lanka, 4 Department of Health Promotion, Rajarata University of Sri Lanka, Mihintale, Anuradhapura, Sri Lanka, 5 Faculty of Nursing, Chiang Mai University, Chiang Mai, Thailand

* prasannah@kdu.ac.lk

**Data Availability Statement:** All relevant data are within the manuscript.

## Abstract

### Background

Cigarette smoking has long been associated with decreased lung function and increased respiratory symptoms globally. While this relationship is well-established, a critical gap exists in our understanding of the specific impact of smoking intensity on individual lung volumes, particularly among the general population. Despite numerous studies conducted on this topic worldwide, there is a noticeable absence of research focusing on the Sri Lankan population, and South Asian studies in this context remain sparse. This study evaluated the prevalence of respiratory symptoms and pulmonary function changes among chronic cigarette smokers and compared them with non-smokers. Furthermore, the proposed study intends to close this gap by undertaking a systematic assessment of the influence of cigarette smoking on lung function and respiratory symptoms in the general population of Sri Lanka. Additionally, the present research represents the first-ever Lung function study conducted in Sri Lanka specifically targeting cigarette smokers.

### Methods

Adult male daily smokers (n = 360) and matched non-smokers (n = 180) from the Colombo district, Sri Lanka, were chosen. Smokers were compared in age, height, and weight with a matched nonsmoking control group. An interviewer provided a questionnaire to collect data on socio-demographic information, smoking behaviors, and clinical respiratory symptoms. Lung function tests were performed with a calibrated PC-based Medikro® Pro (Finland) spirometer and forced vital capacity (FVC), forced expiratory volume in 1 s (FEV1), and FEV1/FVC ratio, peak expiratory flow (PEF), forced expiratory flow between 25% and 75% of FVC (FEF 25–75%) were measured.

**Funding:** The project is financially supported by the University of Sri Jayewardenepura research grant no (ASP/01/RE/MED/2016/55). The funders had no role in study design, data collection and analysis, decision to publish, or preparation of the manuscript.

## Results

Smokers had a significantly higher prevalence of respiratory symptoms and significantly lower FVC, FEV1, FEV1/ FVC, and PEF, FEF 25–75% values than non-smokers. There was a significantly negative correlation of FVC, FEV1, FEV1/ FVC, and PEF, FEF 25–75% with the duration of smoking, and the Brinkman Index. According to the Multiple regression analysis, smoking significantly contributed to deteriorated pulmonary function variables.

## Conclusion

This study revealed that continuous smoking accelerates the deterioration of lung function and increases respiratory symptoms. Early strategies to reduce tobacco use are recommended.

## Introduction

Smoking is a major public health issue in poor and middle-income countries [1, 2] where more than 80% of the world's 1.3 billion tobacco users live, and the impact of tobacco smoking on life expectancy and a person's family economics is enormous [1, 3]. Furthermore, more than 22,000 individuals die each day due to tobacco use, which equates to one person dying every four seconds [4]. Tobacco-related fatalities are estimated to exceed 8.3 million by 2030, with low- and middle-income countries experiencing the greatest increase [5] Tobacco use is the leading cause of chronic obstructive pulmonary disease (COPD) [6, 7] thereby establishing itself as the principal risk factor for compromised respiratory health. [8]. Chronic obstructive pulmonary disease (COPD) manifests in approximately 50% of older individuals who engage in smoking, and it causes 80–90 percent of the morbidity associated with the illness [9, 10]. Although there is no immediate or total cure for COPD, various improvements in health may slow down its progression [11, 12]. Noncommunicable diseases are the leading cause of morbidity and mortality in Sri Lanka, with smoking playing a significant role [13].

In cigarette smoke, particulates, poisons, and reactive compounds cause significant injury to the airway epithelium and its function [14]. Even though many smokers have normal phenotypes, cigarette smoke includes around 1014 oxidant molecules and over 1000 xenobiotics per puff [15], and cigarette smoke exposure produces major physiological alterations in the airway epithelium [16]. These chemicals reduce mucociliary clearance, shorten airway epithelial cilia, and play an essential role in the pathogenesis of smoking-related lung diseases [17]. The fundamental defense mechanisms of cilia are weakened due to the dysregulation of the mucociliary system caused by smoking. This deficiency may cause fluid and mucus collection, necessitating a continuous procedure of airway surface cleansing [14]. These symptoms may suggest the presence of early or ongoing chronic respiratory diseases [18].

A pulmonary function test is an important non-invasive diagnostic test to assess lung health. The pulmonary function test results can be used to prevent or reduce the occurrence of respiratory illnesses by determining the impairment of respiratory function before clinical symptoms appear [19]. Tobacco smoke contains harmful substances that cause inflammation and lung tissue and airway damage. This inflammation can cause airways to narrow, making it harder for air to flow in and out of the lungs efficiently. Smoking might also cause respiratory muscle weakness. Tobacco smoke can cause oxidative stress and inflammation in the respiratory muscles, reducing their strength and endurance. Weakening respiratory muscles

contribute to reduced lung function and increased difficulty in breathing. [8, 20, 21], and is the leading cause of chronic obstructive pulmonary disease [18, 20]. Furthermore, it negatively impacts sedentary smokers more [22]. Some research revealed that quitting smoking increased lung function and life expectancy [23]. Mechanistic studies suggest that the loss in pulmonary function may persist even after quitting smoking [9, 22]. According to the NHLBI Pooled Cohorts Study, previous and low-intensity current smokers lost lung function faster than never-smokers. According to these findings, all levels of smoking exposure are expected to produce long-term and progressive lung damage, and there is no safe level of smoking [20].

Most pulmonary function investigations have focused on FVC, FEV1, and the FEV1/FVC ratio [24]. They discovered that smoking reduced pulmonary function, including forced vital capacity (FVC) [19], forced expiratory volume in one second (FEV1) [18], FEV1/FVC [18], and FEF25-75% [18]. In adult smokers, cigarette smoking induces deficits in FEV1/FVC and FEF25-75, indicating significant airway obstruction and small airway disease [18, 25]. However, a study conducted in Thailand to estimate the prevalence of current smoking and assess pulmonary function among Hill Tribe people with low education and economic status with a high tobacco addiction found no significant differences in pulmonary function compared to non-smoking people from the same tribe [26].

## Methods

### Aim of the study

This study aimed to determine the impact of smoking severity on clinical symptoms and lung function measures in smokers compared to non-smokers. Currently, there has been no comprehensive study conducted to examine the respiratory clinical characteristics of smokers or evaluate lung function among the smoking community in Sri Lanka.

### Study design and setting

A descriptive cross-sectional study was conducted in selected peri-urban MOH (Medical Officer of Health area) divisions of Piliyandala, Homagama, Boralesgamuwa, and Ratmalana in the Colombo district of Sri Lanka.

### Sampling and sample size

The sample size was calculated using the equation of $n = z^2 p (1-p) / d^2$ where n = sample size, z = 1.96; the critical value of specified confidence at 95% confidence interval, p = probable estimate of the proportion of the prevalence of tobacco smoking among males in Sri Lanka (29.9%) (20,21) and d = 5% of absolute error. The minimal sample size was calculated as 330. Also, a 10% sampling error was added to minimize recording errors, and the final sample size was 360. The comparison group (n = 180) was likewise chosen proportionally (2:1). A 2:1 ratio for cases and comparison groups was chosen for study feasibility and cost effectiveness, as well as to provide increased statistical power and flexibility for subgroup analyses while efficiently utilizing [27].

### Inclusion and exclusion criteria

Male tobacco (cigarette) smokers (n = 360) aged between 21–60 years engaged in daily smoking with a history of smoking for a minimum of five years were selected as the study group. Non-smokers (n = 180) were the respondents who affirmed that they had never smoked or not smoked for at least the last five years and had not smoked over 100 cigarettes in their lifetime at the time of recruitment to the study. All participants providing informed written consent

and fulfilling the recruitment criteria were recruited. Smokers with thoracic/abdominal surgery, brain, eye, ear, ENT surgery, pneumothorax, myocardial infarction, ascending aortic aneurysm, hemoptysis, pulmonary embolism, acute diarrhea, angina, severe hypertension (systolic>200 mm Hg, diastolic>120mmHg), Confusion or dementia, discomfort or infection control issues were excluded from the study [28, 29]. Recruitment of subjects was initiated on 15th January 2018 and completed on 01st February 2020.

## Data collection

A pre-tested interviewer-administered questionnaire was used to obtain information onparticipants' baseline data, smoking details, Cardiorespiratory clinical signs.

The Fagerström Test for Nicotine Dependence (FTND) questionnaire was used to assess the degree of nicotine addiction among the participants. FTND is a 6-item scale scored from 0 to 3. The total score is calculated by totaling the responses for the 6 items and a score of 0 to 4 is classified as low dependence, 5 to 7 as moderate, and above 8 is scored as high dependence. A cut-off score of 5 and above was set for nicotine dependence [29]. A study has tested the validity of FTND as a measuring tool in the assessment of smokin cessation programs [30].

The smoking severity was estimated based on the Brinkman Index (BI), which is calculated by multiplying the duration of smoking (in years) by the number of cigarettes smoked per day, [31, 32]. BI is an effective metric for calculating a smoker's lifetime cigarette consumption [33]. The longer and more cigarettes smoked per day increase the value of BI [31, 34].

Information sheets and consent forms were translated into the local languages of Sinhala and Tamil and written informed consent was obtained. Participation was voluntary, and the participants had the right to withdraw from the study at any time. The anthropometric measurements of height and weight were obtained using standardized measurement techniques using a stadiometer (KT-GFO6A-Kindcare- China) and a portable electronic bathroom scale (Omron HN-283-Japan). The individual was advised to stand barefoot on the stadiometer, with heels, hips and shoulders contacting the vertical scale bar, chin straight and inion touching the rear of the vertical scale. The horizontal sliding measure was lowered to the highest point of the head, lightly touching the top. The height was read off the scale to the closest centimeter at the intercept with the sliding measure. All subjects were weighed using an electronic portable bathroom scale with an accuracy of 0.1 kg. The body mass index was calculated using the following formula. BMI $(Kg/m^2)$ = weight (kg)/ [height (m)]$^2$.

Lung function tests were performed for all eligible subjects participating in the study with the PC-based Medikro® Pro (Finland) spirometer embedded with ambient condition measuring sensors for automatic real-time BTPS (body temperature, pressure, water vapour saturated) correction. The sensors of the spirometer tracked environmental conditions automatically and enabled real-time corrections. The results of the three best-breathing manoeuvres were automatically detected and recorded by the Spirometer as well as that result were printed for analysis. South Asian references were selected from the drop-down menu of the PC based Medikro® Pro sofrunningre before run the pirometry test. The Vibration Controlled Tube (VCT) of the spirometer eliminates signal noise caused by the movement of patients and tubes. It stabilizes the signal and improves the overall quality of the spirometry. The results are presented in numerical and graphic form, with further support for the definition of the ventilation function and the diagnosis from the field of spirometry detection. Disposable flow transducers were used to determine the breath volumes and flow-volume curves. The procedure was first demonstrated to participants as a way of explanation and preparation before the actual measurement began. The subject was advised to pay attention to correct posture with the head elevated, complete inhalation, flow transducer position, and complete

exhalation. All maneuver started and finished with resting tidal breathing. It allowed the patient to perform 1–2 sample maneuvers before the measurement. The flow transducer was removed from the mouth after being instructed by the operator. The flow transducer was kept between the teeth to permit the maximal flow to go through the pneumotach and asked to tightly seal with the lips to avoid leakage of air A nose clip was applied to prevent leakage of air from nostrils. Teeth prostheses were removed before the spirometry session. A nose clip was applied during the measurement, three breathing cycles were performed, and the best attempt was selected. Participants with a cough, common cold, inability to perform the test, and other contraindications were not tested. Ventilatory function parameters such as forced vital capacity (FVC), forced expiratory volume in 1 s (FEV1) and FEV1/ FVC ratio, peak expiratory flow (PEF), and forced expiratory flow between 25% and 75% of FVC (FEF $_{25-75\%}$) were measured. All recordings were made in the sitting position. All equipment was calibrated daily and periodically assessed for accuracy. Calibration was done with the standard 3000 ml pulmonary calibration syringe and calibration results were recorded and stored in the calibration log file to ensure uniform calibration of the machine. Disposable SpiroSafe flow transducers provide infection control security. Relevant aseptic measures were followed during the tests. The study was conducted in accordance with the principles outlined by the American Thoracic Society (ATS) [35].

## Statistical analysis

Statistical analysis was performed with IBM SPSS version 23.0 (SPSS Inc.). Frequencies, percentages, and standard deviations were calculated for the main socio-demographic variables and respiratory symptoms. The Chi-square test was used when frequencies were compared. The normal distribution and the similarity of the variances were tested with the Kolmogorov-Smirnov test before statistical analysis. Groups were compared for the parameters with a skewed distribution using the Mann-Whitney U test and expressed as medians (interquartile range). The level of significance was set at $p < 0.05$.

Univariate analysis was done to identify potential significant predictors of the dependent variables ($p < 0.25$) before preparing the Multiple Linear Regression (MLR) model. The smoking or non-smoking status was considered as a predictor variable for standard regression. The Durbin-Watson value was kept between $1.5 < d < 2.5$ to identify first-order linear auto-association in linear regression data. Furthermore, a linear relationship was observed while removing the outliers, and finally, normal distribution was observed among the residuals. A standard multiple regression was run to predict FVC, FEV1, PEF, and FEF 25_75 from smoking status, age, height, and weight.

## Results

### Characteristics of participants

There was no statistical difference ($p > 0.05$) in age, height, weight, and BMI between smokers and non-smokers ($p > 0.05$). (Table 1)

### Frequency of tobacco smoking among smokers

Table 1 depicts the frequency of tobacco smoking among smokers. The majority of the selected population were not heavy smokers. The mean daily consumption of cigarettes by smokers was 5.73±4.88(SD) sticks, the median smoking duration was 21.0±17.0(IQR) years, and the Brinkman index was 80.0±135.0(IQR). The median smoking-initiated age was 18.0 ±4.0 years (Table 2).

**Table 1. Anthropometric characteristics of smokers and non-smokers.**

| Variable | Smokers (n = 360) | | Non-smokers (n = 180) | | p-value |
|---|---|---|---|---|---|
| | Median | IQR | Median | IQR | |
| Age (Years) | 39.0 | 18.75 | 43 | 21.5 | 0.242 |
| Weight (Kg) | 65.0 | 18.97 | 66.0 | 14.5 | 0.652 |
| Height (cm) | 166.5 | 8.00 | 166.0 | 10.3 | 0.412 |
| BMI | 23.24 | 5.92 | 23.50 | 3.67 | 0.738 |

(Mann Whitney U test)

## Respiratory clinical characteristics of smokers and non-smokers

Table 3 depicts the respiratory clinical main features of smokers and non-smokers. There was a significant difference in respiratory health status between smokers and non-smokers, and smokers' clinical characteristics were significantly different compared to non-smokers (p< 0.05) (Table 3).

## Lung function parameters between smokers and non-smokers

Significantly low mean FVC(L), FEV1(L), FEV1/FVC (%), PEF (L/S), and PEF 25–75 (L/S) values of smokers were observed compared to the non-smokers (Table 4).

## Correlation of lung function parameters with smoking variables and anthropometric variables

There was a significant negative correlation between the duration of smoking and FVC, FEV1, FEV1/FVC, PEF, and PEF 25–75 of smokers. Further, Brinkman Index was also significantly and negatively correlated with smokers' FVC, FEV1, FEV1/FVC, PEF, and PEF 25–75.

On the other hand, a statistically non-significant negative correlation was found between the number of cigarettes smoked per day and all measured lung function parameters.

There was a significant negative correlation between the age and FVC, FEV1, FEV1/FVC, PEF, PEF 25–75 of smokers. A non-significant correlation was found between BMI and PEF values.

There was a significant positive correlation between height and FVC; FEV1, PEF 25–75 was also positively correlated (p = 0.001) with the height of smokers. The same association was found with the weight (Table 5).

**Table 2. Frequency of tobacco smoking among smokers.**

| Smoking variable | Smokers (n = 360) | |
|---|---|---|
| | Mean (Median) | SD (IQR) |
| *Number of cigarettes smoked per day | 5.73 | 4.88 |
| Brinkman Index | 80.0 | 135.0 |
| Age of initiation of smoking (years) | 18.0 | 4.0 |
| Duration of smoking (years) | 21.0 | 17.0 |

(*Independent Students t-test & Mann Whitney U test)

**Table 3. Respiratory clinical characteristics of smokers and non-smokers.**

| Variable | | Smokers (n = 360) | Non-smokers (n = 180) | $X^2$ | df | P value |
|---|---|---|---|---|---|---|
| Presence of Cough | | | | | | |
| | Yes | 85(23.7%) | 7(4.2%) | 29.987 | 1 | <0.001 |
| | No | 274 (76.3%) | 160 (95.8%) | | | |
| Character of cough | | | | | | |
| | Phlegm cough | 65(18.1%) | 2(0.3%) | 8.885 | 2 | 0.012 |
| | Dry cough | 20(6.9%) | 0(0%) | | | |
| Frequency of cough | | | | | | |
| | Intermittent | 81(2.5%) | 2(0.6%) | 5.796 | 2 | 0.05 |
| | persistent | 4(1.11%) | 0(0%) | | | |
| Breathing difficulty | | | | | | |
| | Yes | 56(15.6%) | 10(6.25%) | 9.594 | 1 | 0.012 |
| | No | 304(84.4%) | 150(93.75%) | | | |
| Chest tightness | | | | | | |
| | Yes | 40(11.1%) | 5(4.1%) | 10.197 | 2 | 0.006 |
| | No | 320(88.9%) | 165(85.9%) | | | |
| Chest discomfort | | | | | | |
| | Yes | 144(40%) | 149(85.9%) | 67.846 | 3 | <0.001 |
| | No | 216(60%) | 11(4.1%) | | | |

(p-values—Pearson Chi-square test)

## Effect of the smoking factor on lung function parameters (FVC, FEV1, PEF, FEF 25_75)

Univariate analysis was done to identify potential significant predictors of the dependent variables ($p < 0.25$) before preparing the Multiple Linear Regression (MLR) model. The smoking or non-smoking status was considered a predictor variable for standard regression. The Durbin-Watson value was kept between $1.5 < d < 2.5$ to identify first-order linear auto-association in linear regression data. Furthermore, a linear relationship was observed while removing the outliers, and finally, normal distribution was observed among the residuals. A standard multiple regression was run to predict FVC, FEV1, PEF, and FEF 25_75 from smoking status, age, height, and weight (Table 6).

**Table 4. Lung function parameters between smokers and non-smokers.**

| Variable | Smokers (n = 280) | | Non-smokers (n = 131) | | p-value |
|---|---|---|---|---|---|
| | Mean | SD | Mean | SD | |
| FVC (L) | 3.178 | 0.654 | 3.455 | 0.55 | <0.001[#] |
| FEV1(L) | 2.584 | 0.604 | 2.885 | 0.724 | <0.001[#] |
| FEV1/FVC (%) | 80.16 | 8.35 | 82.57 | 7.17 | 0.003[*] |
| PEF (L/S) | 6.631 | 1.696 | 7.60 | 1.707 | <0.001[#] |
| FEF 25–75 (L/S) | 2.664 | 1.098 | 3.032 | 1.074 | 0.002[*] |

(Independent Students t-test)

[#]$p < 0.001$

[*] $P < 0.05$

**Table 5. Correlation of lung function parameters with smoking variables.**

| Variable | FVC (L) (n = 280) | | FEV1(L) (n = 280) | | FeV1/FVC (%) (n = 280) | | PEF(L/S) (n = 280) | | FEF 25-75(L/S) (n = 280) | |
|---|---|---|---|---|---|---|---|---|---|---|
| | r value | p value | r value | p value | r value | p value | r value | p value | r value | p value |
| Duration of smoking (Years) | -0.432 | <0.001# | -0.504 | <0.001# | -0.290 | <0.001# | -0.298 | <0.001# | -0.391 | <0.001# |
| Brinkman Index | -0.199 | <0.001# | -0.252 | <0.001# | -0.148 | 0.017* | -0.183 | 0.003* | -0.221 | <0.001# |
| Num.of cigarettes smoked/day | -0.079 | 0.200 | -0.105 | 0.090 | -0.018 | 0.776 | -0.070 | 0.258 | -0.111 | 0.072 |
| *Age (Years) | -0.466 | <0.001# | -0.539 | <0.001# | -0.291 | <0.001# | 0.282 | <0.001# | -0.403 | <0.001# |

(Pearson correlation and *spearman correlation)

#p<0.001

*p< 0.05

## Effect of the smoking factor on FVC

Multiple regression was run to predict FVC from smoking status, age, height, and weight. Of all the variables, age made the most significant unique contribution ($\beta$ = -0.435), and secondly, smoking ($\beta$ = -0.296). These variables statistically significantly predicted FVC (4,372) = 38.482, p<0.001,0.293. The FVC of smokers reduced by -0.296 L compared to non-smokers due to the effects of smoking. Each one-year increase in age decreased 0.023L of FVC among smokers and non-smokers. When the anthropometry is considered, each 1 cm increase in height increased the FVC by 0.016 L, whereas each 1 kg increased the FVC by 0.017 L.

## Effect of smoking factor on FEV1

Multiple regression was performed to predict FEV1 based on smoking status, age, height, and weight. Of all the variables, age made the most prominent unique contribution ($\beta$ = -0.463),

**Table 6. Effect of the smoking factor on lung function parameters (FVC, FEV1, PEF, FEF 25_75).**

| Variable | Model | Unstandardized coefficient B | Standardized coefficient β | P value | R square | F | P |
|---|---|---|---|---|---|---|---|
| FVC | (constant) | 1.309 | | 0.054 | 0.293 | 38.49 | <0.001 |
| | Smoking status | -0.296 | -0.222 | <0.001 | | | |
| | Age | -0.023 | -0.435 | <0.001 | | | |
| | Height | 0.016 | 0.183 | <0.001 | | | |
| | Weight | 0.005 | 0.107 | 0.023 | | | |
| FEV1 | (constant) | 1.433 | | 0.029 | 0.301 | 40.01 | <0.001 |
| | Smoking status | -0.292 | -0.225 | <0.001 | | | |
| | Age | -0.024 | -0.463 | <0.001 | | | |
| | Height | 0.012 | 0.140 | <0.001 | | | |
| | Weight | 0.006 | 0.115 | 0.023 | | | |
| PEF | (constant) | 3.090 | | 0.121 | 0.204 | 24.9 | <0.001 |
| | Smoking status | -1.005 | -0.272 | <0.001 | | | |
| | Age | -0.045 | -0.307 | <0.001 | | | |
| | Height | 0.029 | 0.118 | 0.018 | | | |
| | Weight | 0.023 | 0.159 | 0.001 | | | |
| **FEF 25_75** | (constant) | 1.952 | | 0.132 | 0.150 | 16.29 | <0.001 |
| | Smoking status | -0.396 | -0.171 | <0.001 | | | |
| | Age | -0.030 | -0.323 | <0.001 | | | |
| | Height | 0.010 | 0.062 | 0.231 | | | |
| | Weight | 0.011 | 0.121 | 0.020 | | | |

and secondly, smoking (β = -0.225). These variables statistically significantly predicted FEV1 (4,372) = 40.01, p<0.001,0.301. FEV1 of smokers reduced by 0.292 L compared to non-smokers due to the effects of smoking. Each one-year increase in age decreased 0.024L of FEV1 among smokers and non-smokers. When the anthropometry is considered, each 1 cm increase in height increased the FEV1 by 0.012 L, whereas each 1 kg increased the FEV1 by 0.006 L.

## Effect of the smoking factor on PEF

Multiple regression was run to predict PEF from smoking status, age, height, and weight. Of all the variables, age made the largest unique contribution (β = -0.307), and secondly, smoking (β = -0.272). These variables statistically significantly predict PEF (4,370) = 24.9, p<0.001,0.204. The PEF of smokers reduced by 1.005L compared to non-smokers due to the effects of smoking. Each one-year increase in age decreases 0.045L of PEF among smokers and non-smokers. When the anthropometry is considered, each 1 cm increase in height increased the PEF by 0.029 L, whereas each 1 kg increased the PEF by 0.023 L.

## Effect of the smoking factor on FEF 25_75

Multiple regression was performed to predict FEF 25_75 based on smoking status, age, height, and weight. Of all the variables, age made the most significant unique contribution (β = -0.23), and smoking (β = -0.171). These variables statistically significantly predicted FEF 25_75 (4,370) = 16.289, p<0.001,0.150. The FEF 25_75 of smokers was reduced by 0.396 L compared to non-smokers due to the effects of smoking. Each one-year increase in age decreased 0.03L of FEF 25_75 among smokers and non-smokers. When the anthropometry is considered, each 1 cm increase in height increased the FEF 25_75 by 0.062 L, whereas each 1 kg increased the FEF 25_75 by 0.121L.

## Discussion

Three hundred sixty daily male tobacco users and 180 non-smokers were randomly recruited for the study. The two groups had no significant difference in age or anthropometric features (p>0.05). The objective of the study was to investigate how cigarette smoking affects lung function and clinical symptoms in male smoking participants from Sri Lanka. The average daily cigarette usage of smokers was not a high frequency compared to many studies. The average daily cigarette usage, smoking duration, and other relevant characteristics were assessed. The age range of our participants was 21–60 years, and this age group was chosen by other studies as well [18, 19].

It is important to recognize certain limitations, despite our best efforts. The study utilised self-reported smoking history and frequency, which could potentially introduce recall bias. The observed effects may have been influenced by the low average daily cigarette usage in our sample. To gain further insights, it would be beneficial to include a more diverse sample with higher smoking intensity.

We examined the smoking severity and compared their clinical characteristics to non-smokers. The Lung function test results of smokers, including FVC, FEV1 in 1 s, FEV1/ FVC ratio, PEF, and FEV between 25% and 75% of FVC (FEF 25–75%) were also compared with non-smokers. Correlation between LFT parameters and smoking duration, Brinkman Index, the number of cigarettes smoked/day, age, BMI, height, and weight with FVC, FEV1, FEV1/ FVC, PEF, and PEF 25–75 were evaluated. Our research has confirmed that cigarette smoking has a negative impact on lung function measures and clinical symptoms in male participants from Sri Lanka. In the research, it was found that symptoms such as cough, breathing difficulties, and chest tightness were commonly reported by smokers. These symptoms were found to

be significantly more prevalent among smokers compared to non-smokers. Smokers reported experiencing an intermittent but frequent expectorant cough, in contrast to non-smokers. Other studies have yielded similar results [36]. A puff of cigarette smoke releases hundreds of trillions of oxidant molecules and thousands of xenobiotics into the [15] respiratory system. These molecules create tremendous stress on airway cilia, change their structure and function, and cause many respiratory ailments [17].

In a study conducted by Boskabady et al., it was observed that smokers in their study experienced significantly higher levels of cough and chest tightness [37]. However, it was observed that the cough type was dry, despite the fact that most participants in our study had a cough with phlegm. Boulet et al. (year) also discovered that smokers frequently experience phlegm coughs, which they identified as an indication of the initial phases of COPD [38]. A study conducted in Finland among young smokers found a significantly higher prevalence of cough and increased sputum production. This study concluded that smoking causes respiratory alterations regardless of ag [39].

Regarding the LFT, smokers had significantly low mean FVC(L), FEV1(L), FEV1/FVC (%), PEF (L/S), and PEF 25–75 (L/S) values compared to the non-smokers. Our results also align with the findings of other studies [24, 39–41]. However, a study conducted by Hasan. et al. found no difference in FEV1 and FVC between smokers and non-smokers, while there was a significant difference in PEFR and FEF 25–75 [42]. Another study conducted in Mexico among adolescents aged between 13–15 years with a shorter smoking history also found a significantly low FEF 25–75 level compared to non-smokers [41].

In our study, there was a significant negative correlation between the duration of smoking and FVC, FEV1, FEV1/FVC, PEF, and PEF 25–75. Rawashdeh & Alnawaiseh also found a significant negative correlation between the duration of smoking with FVC and FEV1 [23, 24]. A study conducted in Thailand among youth smokers revealed no major changes in pulmonary function values with low FVC and no significant change in FEV1 compared to non-smokers. As explained by the authors, the reason was the small duration of smoking of 1–3 years [19].

We did not find a significant relationship between the number of cigarettes smoked daily and lung function parameters. However, we found a non-significant negative correlation with all lung function parameters. Results are compatible with other studies [43]. A pulmonary function study conducted in Jordan among adults ≥ 40 Years old also found a non-significant correlation between the number of cigarettes smoked per day with FVC and FEV1 [23, 24]. However, a cross-sectional multi-center survey of a general population among young adults in Spain [32, 36] found a low FEV1 and a significant association with the number of daily cigarettes. Our participants' mean cigarette consumption was low compared to this study [36], which might be the reason for not showing a relationship with the number of daily cigarettes.

A significant negative correlation existed between the Brinkman Index and lung function measures of smokers. Our findings suggest that more than the number of cigarettes smoked per day, pulmonary function depends on the smoking duration and cumulative smoking exposure, which the Brinkman index or pack-years can measure [8]. They further mentioned that smokers with ten pack-year histories had declined lung function, and another study confirmed no lung changes among those with less than five pack-years [44]. A cross-sectional study conducted in China revealed that smoking causes a decline the pulmonary function; however, effects take time to develop and are not immediately apparent after starting smoking [44]. Our findings also showed that more than the number of cigarettes smoked per day, pulmonary function depends on the smoking duration and cumulative smoking exposure, which the Brinkman index or pack-years can measure.

Advancing age is a reason for decreased lung function, and cigarette smoking duration can accelerate lung function decline, in line with other studies [24]. A cross-sectional study found

that FVC, FEV1, and PEFR were higher in non-smokers in each age group. Age is an independent factor that affects lung functions; however, BMI is not associated with most spirometry values [34, 40]. Rawashdeh & Alnawaiseh also found a significant negative correlation between age, FVC and FEV1. However, the study conducted by Isabel did not find any correlation between anthropometric variables and age [36].

We found a significant positive correlation between height and smokers' lung function parameters, highlighting the potential influence of physical characteristics on respiratory health. However, further research is needed to explore the complex interplay of these factors. According to the results of the Jackson Heart Study Cohort regarding the Joint Effects of Smoking and a Sedentary Lifestyle on Lung Function in African Americans found that non-smokers and physical actives had better lung function, and smokers with a sedentary lifestyle had the lowest lung functions [22].

We found that the FVC of a smoker reduces by 296 ml compared to a non-smoker due to the effects of smoking. According to a cross-sectional study carried out among the Finnish smoking population, there was a 109ml decline in FVC, which is lower than our results [18]. This may be due to other factors such as exercise or ethnic effects. Each one-year increase in age decreases 23ml of FVC among smokers and non-smokers. The decline of FVC is ten times more than the natural decline caused by age.

The FEV1 of smokers reduces by 292 ml compared to non-smokers due to the effects of smoking. A similar level of EEV1 decline was found by Jaakkola et al., wherein the average male, there was a 235 ml reduction in current regular smokers compared to the never smokers [18]. Though the smoking severity was a bit higher than our participants' decline in our study is a bit higher than their results. This may be due to other factors such as exercises or ethnic changes. It was found that each one-year increase in age decreased by 24ml of FEV1 among smokers and non-smokers. The smoking-induced decline of FVC is very significant as it is also ten times higher than a non-smoker.

A smoker's PEF drops by 1005 ml compared to a non-smoker. Smokers and non-smokers both lose 45ml of PEF every year they become older. PEF decline due to smoking is crucial since it is 20 times higher in smokers than non-smokers who experience age-related PEF reduction.

Moreover, the FEF 25_75 of smokers decreases by 396 ml compared to non-smokers. Each one-year increase in age decreased by 30ml of FEF 25_75 among smokers and non-smokers. Jaakkola et al.'s study demonstrated that the decline of FEF 25_75 is 279 ml [18]. This value is significant because it shows the presence of small airway obstruction in a lung function evaluation. The smoking-associated small airway decline indicated by the FEF 25_75 is 13 times higher than its natural decline associated with age. While some research revealed varied results, differences in study design, participant characteristics, and measuring methodology may explain the disparities. The focus of our study on a Sri Lankan male population provides important insights into the influence of smoking on pulmonary function in this specific demographic.

## Conclusion

In conclusion, our research reaffirms the significant adverse effects of cigarette smoking on pulmonary function and respiratory symptoms in male individuals from Sri Lanka. The act of quitting smoking has become recognised as a vital approach to alleviate the rapid decline of lung function that is linked to smoking. Health workers should prioritize the prevention of smoking initiation and the promotion of early cessation, as there is no safe level of smoking for pulmonary function.

## Acknowledgments

We express our deepest gratitude to the participants, medical administrators, and healthcare employees in the Colombo district, the Faculty of Medical Sciences, University of Sri Jayewardenepura, Faculty of Graduate Studies, University of Sri Jayewardenepura, and General Sir John Kotelawala Defence University, Sri Lanka for their continuous support.

## Author Contributions

**Conceptualization:** Prasanna Herath, Savithri Wasundara Wimalasekera, Thamara Dilhani Amarasekara, Manoj Sanjeewa Fernando.

**Data curation:** Prasanna Herath, Savithri Wasundara Wimalasekera, Manoj Sanjeewa Fernando.

**Formal analysis:** Prasanna Herath, Manoj Sanjeewa Fernando.

**Funding acquisition:** Prasanna Herath.

**Investigation:** Prasanna Herath, Manoj Sanjeewa Fernando.

**Methodology:** Prasanna Herath, Savithri Wasundara Wimalasekera, Thamara Dilhani Amarasekara, Manoj Sanjeewa Fernando, Sue Turale.

**Project administration:** Prasanna Herath, Savithri Wasundara Wimalasekera.

**Resources:** Thamara Dilhani Amarasekara, Manoj Sanjeewa Fernando.

**Supervision:** Savithri Wasundara Wimalasekera, Thamara Dilhani Amarasekara, Manoj Sanjeewa Fernando, Sue Turale.

**Validation:** Prasanna Herath, Savithri Wasundara Wimalasekera.

**Visualization:** Prasanna Herath, Savithri Wasundara Wimalasekera.

**Writing – original draft:** Prasanna Herath.

**Writing – review & editing:** Savithri Wasundara Wimalasekera, Thamara Dilhani Amarasekara, Manoj Sanjeewa Fernando, Sue Turale.

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
