## [Decision Letter · Decision Letter 0]

20 Mar 2024

PONE-D-23-31035Comparison of Respiratory Symptoms and Pulmonary Functions of Adult Male Cigarette Smokers and Non-Smokers in Sri Lanka; A Comparative Analytical Study.PLOS ONE

Dear Dr. HERATH,

Thank you for submitting your manuscript to PLOS ONE. After careful consideration, we feel that it has merit but does not fully meet PLOS ONE’s publication criteria as it currently stands. Therefore, we invite you to submit a revised version of the manuscript that addresses the points raised during the review process.

I would like to sincerely apologise for the delay you have incurred with your submission. It has been exceptionally difficult to secure reviewers to evaluate your study. We have now received three completed reviews; the comments are available below. The reviewers have raised significant scientific concerns about the study that need to be addressed in a revision. In particular the reviewers have strong overlapping concerns that you have not provided a clear rationale for your study.  Please revise the manuscript to address all the reviewer's comments in a point-by-point response in order to ensure it is meeting the journal's publication criteria. Please note that the revised manuscript will need to undergo further review, we thus cannot at this point anticipate the outcome of the evaluation process.

We look forward to receiving your revised manuscript.

Kind regards,

Miquel Vall-llosera Camps, Ph.D.

Staff Editor

PLOS ONE

Journal Requirements:

Additional Editor Comments (if provided):

Reviewers' comments:

Reviewer's Responses to Questions

**Comments to the Author**

1. Is the manuscript technically sound, and do the data support the conclusions?

Reviewer #1: Yes

Reviewer #2: Yes

Reviewer #3: Partly

2. Has the statistical analysis been performed appropriately and rigorously? 

Reviewer #1: Yes

Reviewer #2: Yes

Reviewer #3: Yes

3. Have the authors made all data underlying the findings in their manuscript fully available?

Reviewer #1: Yes

Reviewer #2: Yes

Reviewer #3: Yes

4. Is the manuscript presented in an intelligible fashion and written in standard English?

Reviewer #1: Yes

Reviewer #2: Yes

Reviewer #3: No

5. Review Comments to the Author

Reviewer #1: Excessive citations are in introduction. A single statement does not require more than one citation. Use only most recent citations. The tittle and abstract is written well. The methodology is detailed enough. The result is presented step by step. Discussion is adequate.

Reviewer #2: The manuscript addresses a very important public health issue globally. This study is carried out in Th Sri Lanka and compares respiratory health of non-smoking and smoking individuals. Study design is a descriptive cross-sectional study and attempts to fill the existing gap in the research evidence as regards to effect of smoking on pulmonary function in Sri Lanka

The methodology applied in the study is appropriate for the aim of the study. and the research question is clear.

The manuscript would benefit from increased clarity in the methods description and the style of writing overall. Currently the study is described with fairly minimal detail. I do think that adding the actual details, particular related to the methods applied to carry out the study, would make the manuscript stronger. Also I recommend arranging the discussion and specific paragraphs in the more logical way to help to keep message of the manuscript clear.

Reviewer #3: General comments

Authors measured spirometry and symptoms in smokers and non-smokers in Sri Lanka. Smokers had worse pulmonary function and more symptoms.

It is extremely difficult to see what is new with this paper beyond measuring a sample from the Colombo district. If this is the case, then the introduction is far too long and the rationale is not clear from the introduction section.

Paper needs revision for simplicity and clarity. Eliminate all passive voice phrases such as “has been linked to”, “has been shown”. Writing will be clearer if you write in the first person active voice. From the abstract “the current study is expected to” – replace with “We aimed to”

These passive voice issues are throughout the paper.

Specific comments

Page 2 “to the best of the author’s knowledge”. This is an unacceptable phrase. A complete literature search is the responsibility of all authors. From there, an author must own the claim.

Page 5 “to the best of the author’s…”

6. PLOS authors have the option to publish the peer review history of their article (what does this mean?). If published, this will include your full peer review and any attached files.

Reviewer #1: No

Reviewer #2: No

Reviewer #3: No

---

## [Author Response · Author response to Decision Letter 0]

19 Oct 2024

PART 1

Reviewer's Responses to Questions

Comments to the Author

1. Is the manuscript technically sound, and do the data support the conclusions?

Reviewer #1: Yes

Reviewer #2: Yes

Reviewer #3: Partly

2. Has the statistical analysis been performed appropriately and rigorously?

Reviewer #1: Yes

Reviewer #2: Yes

Reviewer #3: Yes

3. Have the authors made all data underlying the findings in their manuscript fully available?

Reviewer #1: Yes

Reviewer #2: Yes

Reviewer #3: Yes

4. Is the manuscript presented in an intelligible fashion and written in standard English?

Reviewer #1: Yes

Reviewer #2: Yes

Reviewer #3: No

5. Review Comments to the Author

Reviewer #1: 

1. Excessive citations are in introduction. 

• Tobacco-related fatalities are estimated to exceed 8.3 million by 2030, with low- and middle-income countries experiencing the greatest increase(5) – Two citations were removed

• These symptoms may suggest the presence of early or ongoing chronic respiratory diseases(17)- Two citations were removed

• Weakening respiratory muscles contribute to reduced lung function and increased difficulty in breathing- Two citations were removed.

• and is the leading cause of chronic obstructive pulmonary disease(17,19).- Two citations were removed

2. A single statement does not require more than one citation. – 8 citations were removed

3. Use only most recent citations. – Done

4. The tittle and abstract is written well. – Accepted with many thanks

5. The methodology is detailed enough. – Accepted with many thanks

6. The result is presented step by step. – Accepted with many thanks

7. Discussion is adequate. – Accepted with many thanks

Reviewer #2: 

1. The manuscript addresses a very important public health issue globally. This study is carried out in Th Sri Lanka and compares respiratory health of non-smoking and smoking individuals. – Accepted with many thanks

2. Study design is a descriptive cross-sectional study and attempts to fill the existing gap in the research evidence as regards to effect of smoking on pulmonary function in Sri Lanka– Accepted with many thanks

3. The methodology applied in the study is appropriate for the aim of the study. and the research question is clear. – Accepted with many thanks

4. The manuscript would benefit from increased clarity in the methods description and the style of writing overall. – Accepted and modified

5. Currently the study is described with fairly minimal detail- Accepted and modified

6. I do think that adding the actual details, particular related to the methods applied to carry out the study, would make the manuscript stronger - Accepted and modified

7. Also I recommend arranging the discussion and specific paragraphs in the more logical way to help to keep message of the manuscript clear - Accepted and modified

Reviewer #3: General comments

1. Authors measured spirometry and symptoms in smokers and non-smokers in Sri Lanka. Smokers had worse pulmonary function and more symptoms. – Yes

2. It is extremely difficult to see what is new with this paper beyond measuring a sample from the Colombo district. If this is the case, then the introduction is far too long and the rationale is not clear from the introduction section.

We appreciate the reviewer's feedback and are pleased to clarify that our study represents the first-ever comprehensive lung function analysis conducted on smokers in Sri Lanka. Our research goes beyond merely measuring a sample from the Colombo district; it contributes valuable insights into the impact of smoking on lung function in a population where such studies are scarce. The regression models employed in our study rigorously identify and quantify lung function decline.

3. Paper needs revision for simplicity and clarity. Eliminate all passive voice phrases such as “has been linked to”, “has been shown”. Writing will be clearer if you write in the first-person active voice. From the abstract “the current study is expected to” – replace with “We aimed to”

- Accepted and appropriately changed. 

4. These passive voice issues are throughout the paper.

- Accepted and most of the issues have been modified

5. Specific comments

Page 2 “to the best of the author’s knowledge”. This is an unacceptable phrase. A complete literature search is the responsibility of all authors. From there, an author must own the claim.

Page 5 “to the best of the author’s…”- 

- Accepted and modified

6. PLOS authors have the option to publish the peer review history of their article (what does this mean?). If published, this will include your full peer review and any attached files.

Do you want your identity to be public for this peer review? For information about this choice, including consent withdrawal, please see our Privacy Policy.

Reviewer #1: No

Reviewer #2: No

Reviewer #3: No

PART II: 

Point by point Responses to Reviewer's Questions

S.No Comment Authors feedback

 Topic 

 Title: Comparison of Respiratory Symptoms and Pulmonary Functions of Adult Male Cigarette Smokers and Non-Smokers in Sri Lanka; A Comparative Analytical Study

The manuscript addresses a very important public health issue globally. This study is carried out in Th Sri Lanka and compares respiratory health of non-smoking and smoking individuals. Study design is a descriptive cross-sectional study and attempts to fill the existing gap in the research evidence as regards to effect of smoking on pulmonary function in Sri Lanka.

The methodology applied int eh study is appropriate for the aim of the study. and the research question is clear. 

The manuscript would benefit from increased clarity in the methods description and the style of writing overall. Currently the study is described with fairly minimal detail. I do think that adding the actual details, particular related to the methods applied to carry out the study, would make the manuscript stronger. Also I recommend arranging the discussion and specific paragraphs in the more logical way to help to keep message of the manuscript clear. 

Specific edits.

I have highlighted in yellow the particular problem within the text clipping and added the comment below to aid in editing work.

 Nothing to correct.

Nothing to correct.

Nothing to correct.

Accepted and changed.

 Introduction:

1) Tobacco use is the leading cause of chronic obstructive pulmonary disease (COPD) (8,9) and the leading risk factor for respiratory health (10).

COMMENT: Unclear what is meant by this. 

2) COPD affects up to 50% of older smokers, and smoking causes 80-90 percent of the morbidity associated with the illness (11,12).

COMMENT: Consider rephrasing this to express the risk group better.

3) Although no rapid or complete cure for COPD exists, early detection is crucial (13,14).

COMMENT: Rephrase to reflect the fact that there is no cure but the progression can slow down with certain health changes

4) Because of the smoking-induced irregular mucociliary system, the cilia's primary defense mechanisms will be impaired, and storing fluids and mucus may result in continuous cleaning of the airway surface (16).

COMMENT: Please clarify, the meaning of this is very unclear.

5) Many studies have indicated that smoking reduces lung volume and respiratory muscle strength, directly and negatively affecting respiratory system function (10,21-23), and is the leading cause of chronic obstructive pulmonary disease (7,22,24,25). 

COMMENT: I don’t understand how smoking affects the respiratory muscle strength. This needs to be clarified so that the real meaning is expressed better.

 Accepted and changed. 

Accepted and changed.

Accepted and changed.

Accepted and changed.

 Clarified in the text

 Methods

Questions:

1. Why choose 2:1 ratio for the case-control groups sizes? Add explanation.

2. Were participants screened for environmental tobacoo smoke exposure? If so what was the criteria applied? I think this should be an important aspect to look at or at least discuss as a potential limitation of the study, if not considered.

6) The anthropometric measurements of height and weight were obtained using standardized measurement techniques using a stadiometer (KT-GFO6AKindcare- China) and a portable electronic bathroom scale (Omron HN-283-Japan).

COMMENT: I would like to read about the accuracy level here too rather than having to go and check the refs. Plse add the details here too.

Accepted and changed.

In addressing the reviewer's comment regarding the absence of assessment for environmental tobacco exposure in our study, we acknowledge that our research primarily focused on active smokers. In our questionnaire, we did include inquiries about environmental tobacco exposure, but due to its relatively low prominence in the raw data sheet and the limited prevalence within the studied population, we did not place explicit emphasis on this aspect during our analysis.

While recognizing that the absence of specific data on environmental tobacco exposure is a limitation of our study, we believe that our primary objective—to investigate the impact of active smoking on pulmonary function and respiratory symptoms—remains valid and insightful. However, we acknowledge that extending our investigation to include environmental exposure could provide a more comprehensive understanding of the overall tobacco-related influences on respiratory health.

Accepted and changed.

 Data collection

COMMENT: Overall in the methods I would like to see more detailed description of the methods. At the moments the methods are not written in the way that they could be followed to perform the work again. Changes and edits related to merely writing style but I do prefer more factual, detailed descriptions. Please consider rewriting the 2 paragraphs of the data collection at least.

1) The Fagerström Test for Nicotine Dependence (FTND) questionnaire

 COMMENT: This is a tool and requires proper referencing. Pls edit.

2) Lung function tests were performed for all eligible subjects participating in the study with the PCbased Medikro® Pro (Finland) spirometer embedded with ambient condition measuring sensors for automatic real-time BTPS (body temperature, pressure, water vapour saturated) correction. The sensors of the spirometer tracked environmental conditions automatically and enabled real-time corrections. The results of the three best-breathing manoeuvres were recorded and printed for analysis. South Asian references were used.

COMMENT: How do you define 3 best breathing manouvers. Pls describe in details. ‘Best’ is unclear, and is different with different measurements. Please add details description.

3) Lung function measurements

COMMENT: Pls add a reference to the source doc where the methods of lung measurements are described.

Accepted and changed. 

Accepted and changed.

Accepted and changed.

Added as suggested

 Results:

Characteristics of participants

1) COMMENT: I would like to see the anthropometrics for male and females too. More informative. Pls consider adding the detail.

Table 1

2) COMMENT: Why was the Brinkman index used. I would like to see the reasoning in brief.

Table 3

3) COMMENT: Why not test for the difference using linear regression where you express the mean difference from the control (Non-smoker). This detail with confidence interval would be much more informative rather than just a cut-off of the Chi-square test.

4) Frequency of Cough. Intermittent/ continues 

COMMENT: I would like to see a description of what these categories mean? And I would use persistent rather than continues as a category name.

Correlation of lung function parameters with smoking variables and anthropometric variables

5) A poor correlation was found between BMI and PEF values and

COMMENT: Pls finish off the sentence. Poor correlation is an unspecific term. Edit and clarify. Use non-significant instead if correlation is non-significant or weak to describe very low but significant correlation.

Table 5: Correlation of lung function parameters with smoking variables

6) COMMENT: Unsure why the anthropometrics are listed in the table. These are already in-built in the lung function measurements. Pls consider removing them. This applies to all tables where lung function parameters are expressed. Pls check all tables and edit accordingly.

 Only males were considered. We do not recruit females as the female smoking prevalence is very low in Sri Lanka

Added to the data collection. 

We appreciate the reviewer's suggestion and acknowledge the potential benefits of utilizing linear regression for mean difference analysis. However, we would like to clarify that our choice of the chi-square test was deliberate and appropriate for the categorical nature of our data.

In our study, we investigated the presence, character, and frequency of cough among smokers and non-smokers, resulting in categorical variables. The chi-square test is specifically designed for analyzing associations between categorical variables, making it a suitable statistical method for our research questions. This test allows us to assess the significance of differences in the distribution of categorical outcomes between the two groups (smokers and non-smokers).

While linear regression is a valuable tool for examining relationships between continuous variables, it may not be the most appropriate choice for our current dataset, as we are dealing with discrete, categorical variables. Moreover, expressing mean differences and confidence intervals for categorical variables may not provide meaningful insights into the nature of our findings.

Intermittent coughing refers to episodes of coughing that occur periodically, with intervals of relief in between. In this category, smokers experience coughing episodes that are not continuous but rather come and go over time.

Continues: The coughing persists without any significant breaks or relief. Individuals in this category who smoke consistently experience a persistent and continuous cough without any noticeable periods of relief.

Changed the word continues as persistent 

Appropriately changed. 

Appropriately changed. Rem

---

## [Decision Letter · Decision Letter 1]

21 Nov 2024

Comparison of Respiratory Symptoms and Pulmonary Functions of Adult Male Cigarette Smokers and Non-Smokers in Sri Lanka; A Comparative Analytical Study.

PONE-D-23-31035R1

Dear Dr. Herath,

We’re pleased to inform you that your manuscript has been judged scientifically suitable for publication and will be formally accepted for publication once it meets all outstanding technical requirements.

Kind regards,

George Kuryan

Academic Editor

PLOS ONE

Additional Editor Comments (optional):

thanks

Reviewers' comments:

Reviewer's Responses to Questions

**Comments to the Author**

1. If the authors have adequately addressed your comments raised in a previous round of review and you feel that this manuscript is now acceptable for publication, you may indicate that here to bypass the “Comments to the Author” section, enter your conflict of interest statement in the “Confidential to Editor” section, and submit your "Accept" recommendation.

Reviewer #3: All comments have been addressed

2. Is the manuscript technically sound, and do the data support the conclusions?

Reviewer #3: Yes

3. Has the statistical analysis been performed appropriately and rigorously? 

Reviewer #3: Yes

4. Have the authors made all data underlying the findings in their manuscript fully available?

Reviewer #3: Yes

5. Is the manuscript presented in an intelligible fashion and written in standard English?

Reviewer #3: Yes

6. Review Comments to the Author

Reviewer #3: Thank you for your revision. I am happy with the changes.

7. PLOS authors have the option to publish the peer review history of their article (what does this mean?). If published, this will include your full peer review and any attached files.

Reviewer #3: No

---

## [Editor Report · Acceptance letter]

28 Nov 2024

PONE-D-23-31035R1 

PLOS ONE

Dear Dr. HERATH, 

I'm pleased to inform you that your manuscript has been deemed suitable for publication in PLOS ONE. Congratulations! Your manuscript is now being handed over to our production team.

Kind regards, 

on behalf of

Dr. PLOS Manuscript Reassignment 

Staff Editor

PLOS ONE